# Exposure to Bacteriophages T4 and M13 Increases Integrin Gene Expression and Impairs Migration of Human PC-3 Prostate Cancer Cells

**DOI:** 10.3390/antibiotics10101202

**Published:** 2021-10-03

**Authors:** Swapnil Ganesh Sanmukh, Nilton J. Santos, Caroline Nascimento Barquilha, Sérgio Alexandre Alcantara dos Santos, Bruno Oliveira Silva Duran, Flávia Karina Delella, Andrei Moroz, Luis Antonio Justulin, Hernandes F. Carvalho, Sérgio Luis Felisbino

**Affiliations:** 1Laboratory of Extracellular Matrix Biology, Department of Structural and Functional Biology, Institute of Biosciences of Botucatu, Sao Paulo State University (UNESP), Botucatu 18618-689, SP, Brazil; ssanmukh@ibecbarcelona.eu (S.G.S.); nilton.unesp@gmail.com (N.J.S.); caroline.barquilha@gmail.com (C.N.B.); sergio.santos@unesp.br (S.A.A.d.S.); flavia.delella@unesp.br (F.K.D.); l.justulin@unesp.br (L.A.J.); 2Laboratory of Extracellular Matrix and Gene Regulation, Department of Structural and Functional Biology, Institute of Biology, State University of Campinas, Campinas 13083-970, SP, Brazil; hern@unicamp.br; 3Department of Histology, Embryology and Cell Biology, Institute of Biological Sciences, Federal University of Goiás (UFG), Goiânia 74690-900, GO, Brazil; brunoduran@ufg.br; 4Laboratory of Monoclonal Antibodies, Department of Clinical Analysis, School of Pharmaceutical Sciences, Sao Paulo State University (UNESP), Araraquara 14800-903, SP, Brazil; andrei.moroz@unesp.br

**Keywords:** bacteriophage, prostate cancer, integrin, PC-3, nanoparticle, cell migration

## Abstract

The interaction between bacteriophages and integrins has been reported in different cancer cell lines, and efforts have been undertaken to understand these interactions in tumor cells along with their possible role in gene alterations, with the aim to develop new cancer therapies. Here, we report that the non-specific interaction of T4 and M13 bacteriophages with human PC-3 cells results in differential migration and varied expression of different integrins. PC-3 tumor cells (at 70% confluence) were exposed to 1 × 10^7^ pfu/mL of either lytic T4 bacteriophage or filamentous M13 bacteriophage. After 24 h of exposure, cells were processed for a histochemical analysis, wound-healing migration assay, and gene expression profile using quantitative real-time PCR (qPCR). qPCR was performed to analyze the expression profiles of integrins *ITGAV*, *ITGA5*, *ITGB1*, *ITGB3*, and *ITGB5*. Our findings revealed that PC-3 cells interacted with T4 and M13 bacteriophages, with significant upregulation of *ITGAV*, *ITGA5*, *ITGB3*, *ITGB5* genes after phage exposure. PC-3 cells also exhibited reduced migration activity when exposed to either T4 or M13 phages. These results suggest that wildtype bacteriophages interact non-specifically with PC-3 cells, thereby modulating the expression of integrin genes and affecting cell migration. Therefore, bacteriophages have future potential applications in anticancer therapies.

## 1. Introduction

Prostate cancer (PCa) is the second most diagnosed cancer in North America [1]. In most cases, owing to its early detection and effective treatment, PCa tends to have good outcomes [2,3]. However, as with most cancers, advanced stages of PCa have limitations with respect to therapy, leading to the necessity for developing targeted treatments [4]. Considering that metastatic castration-resistant or androgen-independent PCa is incurable, there is a need for alternative and effective targeted therapies [5,6,7].

The integrins are important cellular components which are also regulators of matrix metalloproteinases (MMPs). MMPs degrade or remodel the extracellular matrix which can affect various cancer-related processes such as migration, invasion, and metastasis [8,9,10]. Due to the importance of integrins in cancer cell progression, targeting them has been considered as a therapeutic option for most cancers [11,12]. Nanoparticle formulations containing siRNA to silence αv or β1 integrins decreased the phosphorylation of c-Met and of EGFR during liver regeneration, thereby regulating cell proliferation [13,14]. Overexpression of α5 in p53 wildtype liver tumors increased resistance to therapies [15], and the targeting of α5 integrin by miR-26a or inhibiting αvβ3/β5 integrin with cilengitide promoted anoikis [16]. Additionally, inhibition or downregulation of α2β1 integrin diminished the chemoresistance against doxorubicin in leukemia cells [8], and the downregulation of αvβ3 integrin/KRAS and/or β1 integrin in solid tumors was suggested to be useful in EGFR targeted therapies [17,18,19].

In other cancers, such as gastric cancer, FAK signaling is blocked through reduced β1 integrin expression [20]. Most of the malignancies are attributed to hTERT (Human telomerase reverse transcriptase) and, curiously, telomerase is reported to promote cancer invasion in some cancers by enhancing β1 integrin [21,22]. Also, miR-25 regulation of αv had tumor suppressive effects in prostate cancer [23]. The αvβ3, α2β1, and α4β1 integrins are crucial in cancer cell progression to bone metastasis [24]. Moreover, higher expression of α3β1 is associated with a higher biochemical recurrence of prostate cancer [25]. Zheng et al., 1999 [26] reported that the highly invasive PC-3 cell line expresses αvβ3 integrin, whereas the LNCaP cell line does not. Interestingly, the adherence and migration of PC-3 cells on vitronectin, which is an αvβ3 ligand, are possible factors affecting bone metastasis. Additionally, αvβ3 is a mediator of prostate epithelial cell migration through increased tyrosine phosphorylation of focal adhesion kinase (FAK), making it a potential target in prostate cancer cell invasion and metastasis. Increased expression of αvβ3 is also reported to be a good indicator of chemosensitivity [27,28]. Finally, α5β1 integrin, which is a fibronectin receptor, acts as an initiator of angiogenesis [29].

Since most of the integrins recognize the RGD tripeptide motif in extracellular matrix (ECM) components, their strong affinity for complementary bacteriophage coat peptides is reported to aid in internalization by epithelial cells [30,31]. Considering the recent developing interest in both bacteriophages and integrins, it is essential to understand more about their interactions.

Among the different options studied, “Bacteriophages”, which are naturally occurring nanoparticles, have been effectively employed against numerous microbial pathogens, and have been efficiently used for various application purposes in different fields of medicine [32,33,34,35], as they significantly affect our immune system [36,37]. Bacteriophages for targeting cancer cells, both in vitro and in vivo, has emerged as a promising tool since Kantoch and Mordarski demonstrated phage binding to cancer cells [38]. Furthermore, the molecular heterogeneity of the vascular endothelium was demonstrated, as well as the fact that organ targeting is possible using phage peptides [39,40,41,42], which has considerably affected current cancer research and treatment [43,44]. Moreover, modified bacteriophages, including engineered phages and hybrid phages, are considered tremendously important for cancer cell targeting and drug delivery, as reported by independent research groups [45,46,47,48]. Native bacteriophages have been used in the treatment of prostatitis [49]. It is suggested that bacteriophages can reach organs that are usually impermeable to current drugs, which is possibly due to their capacity to cross epithelial and other tissue barriers [50].

In this study, we report the interaction of two different types of bacteriophages, M13 (filamentous phage) and T4 (icosahedral with tail), with human prostate tumor PC-3 cells to understand how their non-specific binding affects cell morphology, cell migration, and the integrin gene expression profile. Considering the importance of integrins in the regulation of important functions associated with cancer cell progression, regulation of associated gene expression related to various pathways, as well as the ECM, we believe that this work will be of significant importance not just to the understanding of how phages interact with prostate cancer cells by affecting their different cellular mechanisms, but also the understanding of the effect of phages on cancer cells during phage therapies, particularly because their use has been broadly considered against bacterial infections in general, and prostatitis in particular.

## 2. Materials and Methods

### 2.1. Cell Culture of PC-3 Cells and Co-Culture with T4/M13 Bacteriophages

The experiment was designed using the PC-3 cell line (ATCC CRL-1435™) obtained from the American Type Culture Collection (Manassas, VA, USA). PC-3 cells were initially cultured in 25 cm^2^ culture flasks (Qiagen, Hilden, Germany) using RPMI1640 medium supplemented with 10% fetal bovine serum (Invitrogen, Carlsbad, CA, USA) and 1% antibiotic/antimycotic solution (*v*/*v*) (Invitrogen, Carlsbad, CA, USA). Cells were maintained under strict sterile conditions inside a 5% CO_2_ incubator (Thermo Scientific, Waltham, MA, USA) at 37°C, and were later transferred to 75 cm^2^ culture flasks (Qiagen, Hilden, Germany). After obtaining the necessary cells for the experiments, PC-3 cells were re-seeded at 2 × 10^4^ cells/cm^2^ for each well in a 12- or 6-well cell culture plate. All the experiments were performed in triplicate as per standard procedures and guidelines provided by respective authorities.

Bacteriophages M13KE (New England Biolabs Inc.—Ipswich, MA, USA) and Coliphage T4 (T4r+) (Carolina Biological Supply Co., Burlington, NC, USA) were purchased pre-purified and processed as per previously published protocols [31] with some modifications. To avoid any contamination, as well as any other issues associated with the bacterial debris (e.g., lipopolysaccharides (LPS) and endotoxins), phages were never expanded in bacterial culture. The original stock obtained from the manufacturer, each with 10^11^ pfu/mL, was centrifuged at 10,000× *g* for 30 min followed by filtration through a 0.22 μm cellulose acetate membrane filter (Merck Millipore™, Burlington, MA, USA). The supernatants were collected with sterile tips in sterile falcon tubes and were labelled as original phage stocks. The 100X dilutions were prepared for each phage in phosphate buffer saline (PBS) and labelled as a working stock solution with a concentration of 1 × 10^9^ pfu/mL. The phages were further diluted in cell culture medium to reach 1 × 10^7^ pfu/mL and 1 × 10^5^ pfu/mL for treating the cells. This series of dilutions and processing significantly reduced the peptone broth components in the cell culture medium and any possible trace of LPS or endotoxin. 

### 2.2. Morphological Investigation

The PC-3 cells were grown in 12-well plates containing sterile coverslips on the bottom for morphological analysis. After reaching 30% confluency, PC-3 cells were exposed to 10^7^ pfu/mL (10 μL of 10^9^ stock solution in 1 mL of culture medium) of bacteriophage M13 or T4 treatment for 24 h. PC-3 cells treated with 10 μL of PBS were used as negative control. Histochemical analysis was performed to verify how T4 and M13 phages affected PC-3 cell morphology. The coverslips were washed in PBS and fixed with 10% formaldehyde in PBS for 30 min, and then washed. Cells were stained by hematoxylin-eosin, dried in ethanol, mounted in a glass slide with Permount, and observed in a Leica DMLB conventional light microscope (Leica Inc., Wetzlar, Germany). Cells with spindle-shaped morphologies were counted in 5 different fields per well in triplicate and in three independent experiments. The counting results were expressed as the percentage of total cells counted.

### 2.3. Wound Healing Assay

PC-3 cell migration studies were performed using 12-well plates with and without treatment with T4 and M13 bacteriophages. Initially, PC-3 cells were cultured in 12-well plates at 1 × 10^5^ cells/cm^2^ until 100% confluence was achieved. Then, the monolayer was scratched across the well in a straight line using a 100–200-µL pipette tip. After scratching, each well of the 12-well plates was washed twice with D-PBS (Invitrogen) to remove detached cells and cell debris, and fresh RPMI 1640 medium was added to each well. Then, T4 or M13 bacteriophages were added to the respective wells at 1 × 10^5^ pfu/mL (10 μL from a 1 × 10^7^ stock solution). In a previous work, we tested phages at 1 × 10^7^ for 48 h [51], and now we would like to use a lower concentration for 72 h. For T4 and M13 bacteriophage treatments, each of the three wells was inoculated with phage culture medium. At 0 and 72 h of culture, images of healing areas were obtained from each control and treatment using an inverted phase-contrast microscope with a digital camera. ImageJ was used to measure the empty area. The empty area of treated cells was presented as a percentage of the original 100% empty area.

### 2.4. Extraction of RNA and cDNA Synthesis for qPCR Profiling

PC-3 cells were cultured in 6-well plates at 1 × 10^4^ cells/cm^2^ until 70% of confluency was achieved. Afterwards, cells were treated with T4 and M13 bacteriophages at 1 × 10^7^ pfu/mL for 24 h. Other cells treated with 10 μL of PBS in 1 mL of culture medium were used as controls. After 24 h of treatment, cells were washed in PBS and submitted for total RNA extraction using the All-Prep DNA/RNA/Protein extraction kit according to the manufacturer’s instructions (Qiagen, Hilden, Germany). Total RNA was then quantified using a NanoVue Plus spectrophotometer (GE Healthcare, Barrington, IL, USA), which also allowed for an estimation of RNA purity by measuring absorbance at 260 nm (RNA quantity) and 280 nm (protein quantity). A ratio between 1.8 and 2.0 is considered to be a marker for high quality and purity of the samples. RNA reverse transcription was performed using a High-Capacity cDNA Archive Kit (Thermo Scientific, Waltham, MA, USA) according to the manufacturer’s guidelines.

The mRNA expression levels were measured by qPCR using the QuantStudio^TM^ 12K Flex Real-Time PCR System (Thermo Scientific, Waltham, MA, USA). The qPCR performed was compliant with the Minimum Information for Publication of Quantitative Real-Time PCR experiment guidelines. cDNA samples were amplified using SYBR^®®^ Green Master Mix (Thermo Scientific, Waltham, MA, USA), and the specific primers were synthesized and acquired from Invitrogen. The reactions were performed at 95 °C for 10 min, followed by 40 cycles of denaturation at 95 °C/15 s, and annealing/extension at 60 °C for 1 min. The specificity of each primer was evaluated by the dissociation curve at the end of each PCR reaction, which confirmed the presence of a single fluorescence peak. The relative quantification of expression was performed using the 2^−∆∆Ct^ method [52] using the DataAssist v3.01 software (Thermo Scientific, Waltham, MA, USA).

These reactions were also performed in triplicate for the target genes: *ITGA5*, *ITGAV*, *ITGB1*, *ITGB3,* and *ITGB5*, along with endogenous *ACTB* controls. According to the expression stability among all samples, the reference gene *ACTB* was used to normalize mRNA expression. Forward and reverse primers are listed in Table 1.

### 2.5. Statistical Analysis

Statistical analyses were performed using the GraphPad Prism software (version 5.00, Graph Pad, Inc., San Diego, CA, USA). The results were analyzed using analysis of variance, followed by the Tukey–Kramer test, and the data are expressed as mean ± the standard deviation (SD). Differences were considered statistically significant when *p* was <0.05.

## 3. Results

### 3.1. Bacteriophage-Treated PC-3 Cells Adopt a Spindle-Like Phenotype

There was a significant shift in morphology in treated cells toward a more spindle-shaped morphology (Figure 1A). Spindle-shaped cells were counted and divided by the total number of cells per microscopic field. The number of spindle-shaped cells was significantly higher in the bacteriophage-treated cells (Figure 1B). 

### 3.2. Bacteriophage-Treated PC-3 Cells Have Compromised Migration

The effect of the interaction of the PC-3 cell line with T4 and M13 bacteriophages was also evaluated by performing the scratch wound healing assay for 72 h. Both phages slowed down the closure of the wounded area significantly. The M13 bacteriophage was more effective in migration inhibition (80.88% of wound coverage) than the T4 bacteriophage (84.24% of wound coverage) (Figure 2).

### 3.3. Bacteriophage-Treated PC-3 Cells Have Increased Integrin Gene Expression

Exposure to T4 and M13 bacteriophages for 24 h induced a significant increase in mRNA expression of integrin genes. The relative mRNA expressions of *ITGAV*, *ITGA5*, *ITGB3*, and *ITGB5* were significantly upregulated (Figure 3).

## 4. Discussion

Bacteriophages represent an important tool for exploring tumor cell nanoparticle interactions and drug delivery, and their use was promptly verified based on the results of Ivarsson et al. [53], who developed phage libraries which map several human and viral peptides. Using this biotechnological approach, new short linear motifs, which can be used as docking sites for protein domains, were successfully determined for numerous proteins. The authors also conclude that this is an interesting method of investigating host–pathogen and protein–protein interactions, which has been proven to be true in recent years. Native bacteriophages might also have additional therapeutic functions, such as in the treatment of prostatitis [49], due to their capacity to cross epithelial barriers [50].

Here we show that PC-3 cells and the bacteriophages T4 and M13 interactions are consistent with the results reported in previous studies reported by us as well as other groups [37,51,54,55,56]. Our major findings can be summarized as follows. (i) After treatment with the phages, a robust shift in morphology was observed, with a higher percentage of more spread-out spindle-shaped cells. (ii) The migration assay revealed that the interactions between the phages and cells significantly increased the time required for wound closure, which points to the effect of reducing cell migration; this effect was also observed in melanoma cells treated with these phages [56] and our recently reported studies [37,51,54]. (iii) The expression of several integrin genes was upregulated, and a possible physical interaction between phage peptides and integrins from the cells via RGD peptide sequences should not be ruled out [57,58,59,60,61].

However, the effects of these interactions are conflicting; as previous authors have reported, they could promote more adhesion affinity in cancer cells, impairing their growth and migration [56]. Yet, the effects of integrin binding and upregulation can be either positive or negative factors of cell migration [54,55,57,58,59], but in our recently reported s-ARU (semi-adherent relative upsurge) method, which is a modification of the gap closure method, we observed that bacteriophages M13 and T4 significantly affected the migration of PC-3 and LNCaP cancer cell lines in vitro [51]. Similarly, we also observed that these phages significantly downregulated the *HSP90* gene, following binding to PC-3 cell lines, which is reported to be overexpressed in cancer cells and is responsible for expressing antiapoptotic proteins [54]. Considering the androgen-independent nature of PC-3 cells, which were derived from grade IV adenocarcinoma, highly metastatic, osteolytic and tumorigenic [62,63], the importance of our findings cannot be sidelined. It is also possible that the observed effects are due to bacteriophage interactions with the beta-3 integrin, possibly via the KGD peptide present in the bacteriophage, as reported before [57,58,59,60,61].

The upregulation of *ITGAV* and *ITGB3* can make the cancer cells resistant to chemotherapy, as reported previously by different groups [27,28]. Overexpression of these integrins upon phage interaction confirms the availability on both phages of a natural antagonist for RGD, which is reported to help cancer cells survive following ionizing radiation (IR) through survivin regulation [3,37]. In ovarian and breast cancers, it has been reported that overexpression of *ITGAV* and *ITGB3* promotes cancer pro-survival pathways by increased expression of BCL-2 (B-cell leukemia/lymphoma 2) antiapoptotic family proteins and triggering PI3-K/Akt and Ras-Raf-ERK/MAPK pathways for cancer cell proliferation and survival from chemotherapy and/or ionizing radiation [27]. Similarly, in oral squamous cell carcinoma, overexpression of *ITGAV* is reported to activate MEK/ERK signaling pathways promoting proliferation and invasion [64]. Additionally, it was observed that the overexpression of integrin *ITGA5* was associated with lymph node metastasis and tumor size growth in esophageal squamous cell carcinoma [65]. Significant overexpression of *ITGB5* has been linked with progressive malignancies in glioblastoma, even in patients undergoing radiotherapies. Along with this, *ITGB5* not only promotes migration but also invasion, and has been reported to affect immune response and angiogenesis within the tumor microenvironment [66]. Our results suggest that “phage-induced” integrin overexpression by PC-3 cells could make them more susceptible to this druggable pathway, such as AKT/PI3-K/MAPK/ERK and FAK/ERK/MAPK inhibitors [20]. Further in vitro and in vivo studies are needed to assess its effectiveness in a controlled manner.

The recent development in the field of bacteriophage research and application shows that bacteriophages are capable of utilizing the route of mammalian viruses for entry into mammalian cells and they can be detected in both the cytoplasm and the cell nucleus, affecting various cellular processes along with retaining their lytic activity towards their bacterial hosts [67,68,69,70]. The size-dependent uptake and degradation of viral particles has been reported [71,72,73] and must be of considerable implication in non-specific targeting of cancer cells. Similarly, the integrins which are responsible for cellular behavior were significantly modulated by phages, therefore their role in modulating cancer progression genes cannot be ignored [74,75,76]. Our study supports the “Bacteriophage” as a model nanoparticle for in-vivo cancer studies along with its therapeutic potential [77].

## 5. Conclusions

In conclusion, we showed that natural bacteriophages interact non-specifically with PCa cells, which triggers internal cellular signaling and gene expression alterations. The mechanisms underlying these effects remain unknown. However, the effects observed in many genes demonstrate that phages have a broad range of binding properties via their proteinaceous external coat, which may be useful as natural peptide displays for drug delivery and anticancer therapies. Considering recent reports of phage internalization, novel cancer therapeutic applications are possible through bacteriophage-based peptide/drug delivery and/or by the synergistic effect of phage therapy and integrin pathway inhibitors.

## Figures and Tables

**Figure 1 antibiotics-10-01202-f001:**
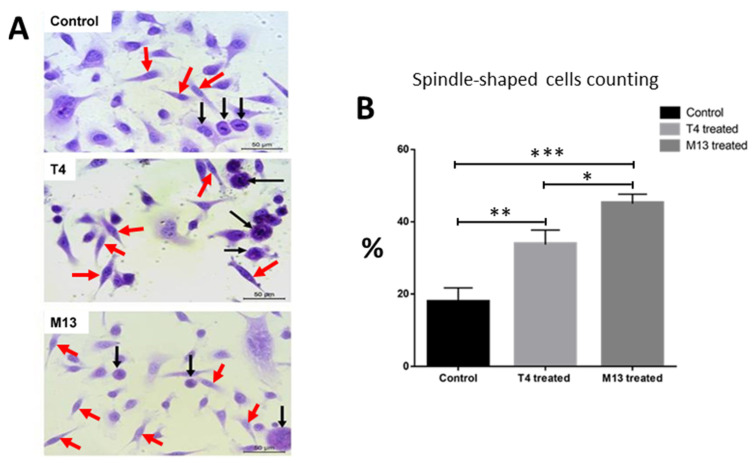
(**A**) Representative images of prostate tumor cells PC-3 cultivated on coverslips and stained with hematoxylin and eosin, which were treated for 24 h with 1 × 10^7^ pfu/mL of either lytic T4 bacteriophage or filamentous M13 bacteriophage along with control. Untreated PC-3 cells (control); PC-3 cells exposed to the T4 bacteriophage; PC-3 cells exposed to the M13 bacteriophage. Note the presence of spindle-shaped cells (red arrows) in the phage-treated groups in contrast with round-shaped cells (black arrows). Scale bars = 100 µm; (**B**) spindle-like cells on the coverslips were counted per field in 10 different fields per well. Compared to control (vehicle-treated cells) T4-treated cells and M13-treated cells had a statistically significant higher number of cells with spindle shaped morphology with *p* < 0.01 (**) and *p* < 0.001 (***), respectively. M13-treated cells has also statistically significant higher number of cells with spindle-shaped morphologies with *p* < 0.05 (*).

**Figure 2 antibiotics-10-01202-f002:**
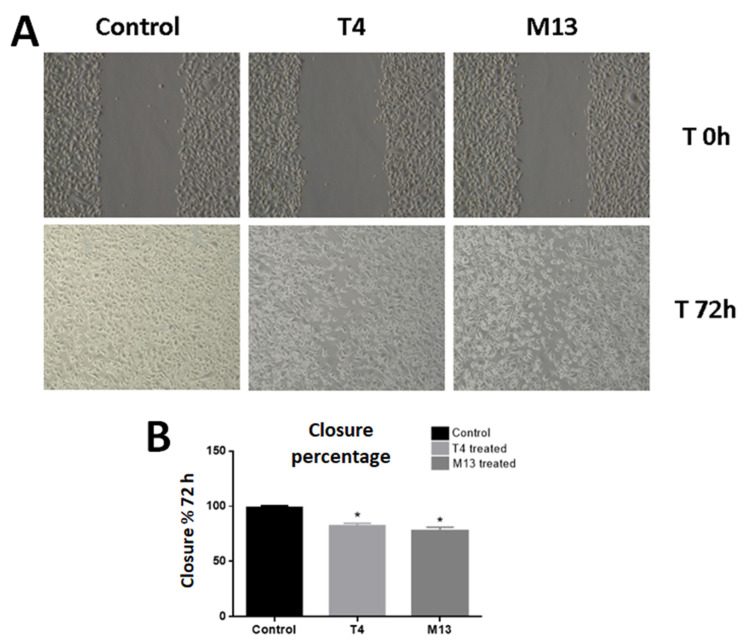
(**A**) Representative images of the wound healing assay performed for the PC-3 cells, which show that migration is affected by both T4 and M13 phages. Cells were treated with 1 × 10^5^ pfu/mL concentration of either lytic T4 bacteriophage or filamentous M13 bacteriophage or with 10 μL of PBS (control). (**B**) Measurement of empty areas confirmed that after 72 h, both phage treatments impaired the migration of PC-3 cells. Asterisks mean statistical difference related to the control group with *p* < 0.01.

**Figure 3 antibiotics-10-01202-f003:**
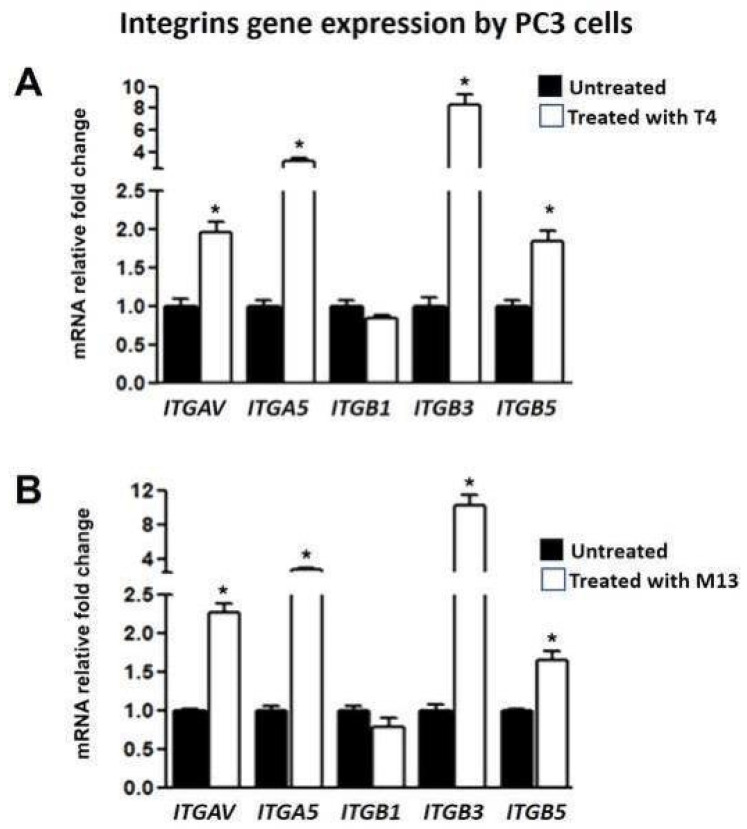
Integrin (*ITGAV*, *ITGA5*, *ITGB3*, and *ITGB5*) gene expression analysis in the PC-3 cell line after their interaction with T4 (**A**) and M13 (**B**) bacteriophages. Genes were significantly overexpressed by both phages. Expressions were normalized with respect to *ACTB* expression. Asterisks mean *p* < 0.01, as compared to the control.

**Table 1 antibiotics-10-01202-t001:** The list of RT-qPCR primers used for studying different integrin gene expressions.

Genes	Sense Primer	Antisense Primer
*ACTB*	GATTCCTATGTGGGCGACGA	TGTAGAAGGTGTGGTGCCAG
*ITGA5*	GGGTGGTGCTGTCTACCTC	GTGGAGCGCATGCCAAGATG
*ITGAV*	AGGCACCCTCCTTCTGATCC	CTTGGCATAATCTCTATTGCCTGT
*ITGB1*	GCCAAATGGGACACGCAAGA	GTGTTGTGGGATTTGCACGG
*ITGB3*	CTGCCGTGACGAGATTGAGT	CCTTGGGACACTCTGGCTCT
*ITGB5*	GGGCTCTACTCAGTGGTTTCG	GGCTTCCGAAGTCCTCTTTG

## Data Availability

Data is contained within the article.

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
