# Peer review of "Exposure to Bacteriophages T4 and M13 Increases Integrin Gene Expression and Impairs Migration of Human PC-3 Prostate Cancer Cells"

_antibiotics, 2021, doi:10.3390/antibiotics10101202_

Round 1
Reviewer 1 Report
- In the beginning of the abstract the cell line is denoted PC-3, thereafter PC3.
- P3 line 135: The authors claim that only samples with 260/280 ratio less or equal to 1.8 were analyzed – should it not be the other way around? Usually, a ratio between 1.8-2.0 is considered to be a marker for high quality and purity of the samples.
- Treatment with the phages renders a spindle cell morphology – why is this? It would be nice if the authors could elaborate of what this change in morphology tells about what is happening with the cell – what characteristics are connected to this morphology?
- P8 line 215: The authors claim that the prolonged wound closure time is a result of both reduced cell proliferation and migration – how can the authors claim that the proliferation rate is reduced?
Author Response
Response to reviewer’s comments:Reviewer 1:
- In the beginning of the abstract the cell line is denoted PC-3, thereafter PC3.
Response: The corrections were made as per reviewer’s suggestion throughout the article.
- P3 line 135: The authors claim that only samples with 260/280 ratio less or equal to 1.8 were analyzed – should it not be the other way around? Usually, a ratio between 1.8-2.0 is considered to be a marker for high quality and purity of the samples.
Response: We are thankful for the reviewer’s minute observation. The corrections were made as per reviewer’s suggestion in the article.
- Treatment with the phages renders a spindle cell morphology – why is this? It would be nice if the authors could elaborate on what this change in morphology tells about what is happening with the cell – what characteristics are connected to this morphology?
Response: We have provided more information explaining the spindle-shaped cell morphology and explained why this morphology is likely related to phage interaction.
- P8 line 215: The authors claim that the prolonged wound closure time is a result of both reduced cell proliferation and migration – how can the authors claim that the proliferation rate is reduced?
Response: This is an interesting question. We already have observed this effect of phage treatment in other cell line LNCaP (Sanmukh et al., Bacteriophages M13 and T4 Increase the Expression of Anchorage-Dependent Survival Pathway Genes and Down Regulate Androgen Receptor Expression in LNCaP Prostate Cell Line. Viruses. 2021; 13(9):1754. https://doi.org/10.3390/v13091754). We believe that increased expression of integrins leads to a more attached cell and to a less proliferative behavior. We also have observed this behavior for PC-3 cells in a 100 times higher concentration of phages (Sanmukh SG, Felisbino SL. Development of pipette tip gap closure migration assay (s-ARU method) for studying semi-adherent cell lines. Cytotechnology. 2018 Dec;70(6):1685-1695. doi: 10.1007/s10616-018-0245-1.)

Reviewer 2 Report
The manuscript by Sanmukh et al., analyzed the migratory behavior of human PC3 prostate cancer cells by treatment with two types of bacteriophages. Since bacteriophages might be used as therapeutics the effect on tumor cells is not yet clear. Authors observed a reduced migration. Associated was an increase of mRNA expression of some integrins.
In general, the amount of novelty is very limited. The manuscript lacks analysis of pathways and specificity of treatment. The change of mRNA expression of some integrins is only an association.
Major points:
- Unclear is whether the effect on migration by 10 Mio Pfu per ml of viruses, which possess coated proteins, is specific? Or, would any other protein at that concentration show a similar effect?
The important question is whether the high doses of viruses can be replaced by just adding an unspeficic protein, such as albumin or others? Authors shall reveal the virus-specific effect by treatment with one unrelated protein at high doses.
- Authors must analyze which intracellular factors of the inhibited migration are affected. Factors such as FAK, p-FAK, N-, and C-cadherins, MET, vimentin and others will provide insights into the bacteriophage- induced changes.
Author Response
Response to reviewer’s comments:
Reviewer 2:
The manuscript by Sanmukh et al., analyzed the migratory behavior of human PC3 prostate cancer cells by treatment with two types of bacteriophages. Since bacteriophages might be used as therapeutics the effect on tumor cells is not yet clear. Authors observed a reduced migration. Associated was an increase of mRNA expression of some integrins.
In general, the amount of novelty is very limited. The manuscript lacks analysis of pathways and specificity of treatment. The change of mRNA expression of some integrins is only an association.
Response: The authors thank the observations of the reviewer. We have included new references and literature supporting our work in the introduction section and have also supported it with our previously reported work to explain the significance of the present findings. References were included to explain the importance of integrin mediated pathways co-regulating gene expressions associated with other pathways involved in cancer cell progression.
Major points:
- Unclear is whether the effect on migration by 10 Mio Pfu per ml of viruses, which possess coated proteins, is specific? Or would any other protein at that concentration show a similar effect?
Response: This is an interesting observation. As explained previously, a time and concentration dependent experiment were performed for LNCaP cancer cell lines (Sanmukh and Felisbino, 2018; Sanmukh et al., 2021). We and other authors have found robust and specific induction of gene protein expression by phage interaction. Furthermore, our group and other researchers use culture conditions from 1 to 10% of Bovine fetal serum, which contains several globulins and albumins, and such integrin gene expression modulations are not observed. So, we believe that integrin overexpression observed in our study is due to phage interaction.
- The important question is whether the high doses of viruses can be replaced by just adding an unspecified protein, such as albumin or others? Authors shall reveal the virus-specific effect by treatment with one unrelated protein at high doses.
Response: As explained previously, we believe that integrin overexpression observed in our study is due to phage interaction. But future studies from our lab certainly will address this issue.
- Authors must analyze which intracellular factors of the inhibited migration are affected. Factors such as FAK, p-FAK, N-, and C-cadherins, MET, vimentin and others will provide insights into the bacteriophage- induced changes.
Response: This is a very interesting question. The authors agree with the reviewer that this information is very relevant. However, such analysis is not feasible at this moment. As suggested by the reviewer, we have included the importance of integrin mediated pathways in the introduction and their role in regulating other cancer progression pathways in detail. These observations go in line with the previously reported studies from our group (Sanmukh etal., 2017; Sanmukh and Felisbino, 2018; Sanmukh et al., 2021).
Cheng, Y. J., Zhu, Z. X., Zhou, J. S., Hu, Z. Q., Zhang, J. P., Cai, Q. P., & Wang, L. H. (2015). Silencing profilin-1 inhibits gastric cancer progression via integrin β1/focal adhesion kinase pathway modulation. World journal of gastroenterology, 21(8), 2323–2335. https://doi.org/10.3748/wjg.v21.i8.2323.
Vellon, L., Menendez, J. A., Liu, H., & Lupu, R. (2007). Up-regulation of alphavbeta3 integrin expression is a novel molecular response to chemotherapy-induced cell damage in a heregulin-dependent manner. Differentiation; research in biological diversity, 75(9), 819–830. https://doi.org/10.1111/j.1432-0436.2007.00241.x.
Zhao, X., & Guan, J. L. (2011). Focal adhesion kinase and its signaling pathways in cell migration and angiogenesis. Advanced drug delivery reviews, 63(8), 610–615. https://doi.org/10.1016/j.addr.2010.11.001.

Round 2
Reviewer 1 Report
I think that it is better to publish this paper as a short communication than a full length research article, so I have no objections to this.
Unfortunately, the authors have misunderstood my comment no 4 regarding effects on both proliferation and migration. In the cover letter they have commented on why this could be, which is very interesting to read. But my objection is that they claim that they have an effect on both those properties without showing any evidence for it in the work they present. What they have shown is that the wound closure is affected, but they do not do any experiments to investigate if this is only due to a change in migration of the cells, or if it is a combined effect of both proliferation and migration. My objection to their interpretation is that they have not measured the proliferation rate of the cells so they have no idea if this is affected or not. Or if they have, they do not present those results in the paper. In my mind they draw conclusions from their results that are not valid in this respect.
Author Response
Response to reviewer’s comments:
Reviewer 1 comment:
I think that it is better to publish this paper as a short communication than a full-length research article, so I have no objections to this.
Unfortunately, the authors have misunderstood my comment no 4 regarding effects on both proliferation and migration. In the cover letter they have commented on why this could be, which is very interesting to read. But my objection is that they claim that they influence both those properties without showing any evidence for it in the work they present. What they have shown is that the wound closure is affected, but they do not do any experiments to investigate if this is only due to a change in migration of the cells, or if it is a combined effect of both proliferation and migration. My objection to their interpretation is that they have not measured the proliferation rate of the cells so they have no idea if this is affected or not. Or if they have, they do not present those results in the paper. In my mind they draw conclusions from their results that are not valid in this respect.
Response:
We agree with the observation of the reviewer. We do not have additional experiments about cell proliferation to claim this effect, so we decided to remove this statement from the manuscript. In fact, the title of our manuscript points to an impairment in cellular migration process, observed in the wound healing experiment. And we believe this impairment has a relationship with integrin over-expression and signaling, induced by phage exposure. As suggested by the reviewer we are resubmitting the manuscript as a short communication.
Reviewer 2 Report
In general, the amount of novelty is still very limited. The manuscript lacks analysis of pathways and specificity of treatment. The change of mRNA expression of some integrins is only an association.
Author Response
Response to reviewer’s comments:
Reviewer 2 comment:
In general, the amount of novelty is still very limited. The manuscript lacks analysis of pathways and specificity of treatment. The change of mRNA expression of some integrins is only an association.
Response:
Considering the limitations associated with our reported studies, we are resubmitting our manuscript as a short communication. Although limited, we believe that any additional data or publication are very important in this field to shed light on the bacteriophage research applications. Those integrin pathways signaling will certainly be an issue for future works in our research group.
Round 3
Reviewer 1 Report
The description of the statistics in Fig 1 could improve - unfortunately I do not understand it as it is stated at the moment. I would prefer a description similar to what can be found in Fig 2 and 3. If I should guess, I think there is a significant difference control vs T4 and control vs M13 that is p<0.01, but is there also a significant difference if you compare the effect of T4 vs M13?
Author Response
The authors thanks the reviewer for the observation. There is a significant difference with p < 0.05 between T4 and M13 effec and this information was missing in the figure. We have ajusted the plot as suggested and replaced the figure.
Reviewer 2 Report
One comment to tune down the statment ob page 11 line 297:
Please rather state that: "from our results, it suggests that "phage-induced" integrin.....
instead of "it is clear" that "phage-induced"
Author Response
The authors thank the reviewer for the careful revision of our manuscript and for the important contributions to improve the quality of it. The sentence has been changed as suggested.